# Cancer History Avoids the Increase of Senescence Markers in Peripheral Cells of Amnestic Mild Cognitive Impaired Patients

**DOI:** 10.3390/ijms24087364

**Published:** 2023-04-17

**Authors:** Carol D. SanMartín, Felipe Salech, Daniela Paz Ponce, Jorge Concha-Cerda, Esteban Romero-Hernández, Gianella Liabeuf, Nicole K. Rogers, Paola Murgas, Bárbara Bruna, Jamileth More, María I. Behrens

**Affiliations:** 1Centro de Investigación Clínica Avanzada (CICA), Facultad de Medicina-Hospital Clínico, Universidad de Chile, Independencia, Santiago 8380453, Chile; dazilsanmartin@gmail.com (C.D.S.); fhsalech@gmail.com (F.S.); dponcedelavega@gmail.com (D.P.P.); jconchacerda@gmail.com (J.C.-C.); bbruna@ug.uchile.cl (B.B.); jamilethmore@gmail.com (J.M.); 2Departamento de Neurología y Neurocirugía, Hospital Clínico Universidad de Chile, Independencia, Santiago 8380430, Chile; 3Programa de Fisiología y Biofísica, Instituto de Ciencias Biomédicas (ICBM), Facultad de Medicina, Universidad de Chile, Santiago 8380453, Chile; esteban.romero94@gmail.com; 4Laboratorio de Obesidad y Metabolismo Energético (OMEGA), Instituto de Nutrición y Tecnología de los Alimentos (INTA), Universidad de Chile, Santiago 7830490, Chile; gianella.alejandra@gmail.com; 5Departamento de Neurociencia, Facultad de Medicina, Universidad de Chile, Independencia, Santiago 8380453, Chile; n.rogersc@gmail.com; 6Instituto de Bioquímica y Microbiología, Facultad de Ciencias, Universidad Austral de Chile, Valdivia 5110566, Chile; paolamurgas@gmail.com; 7Laboratorio de Psiquiatría Traslacional, Departamento de Psiquiatría Norte, Universidad de Chile, Santiago 8380539, Chile; 8Departamento de Neurología y Psiquiatría, Clínica Alemana-Universidad del Desarrollo, Santiago 8370065, Chile

**Keywords:** cellular senescence, cancer history, amnestic mild cognitive impairment (aMCI), peripheral blood mononuclear cells (PBMCs), Alzheimer’s Disease (AD), aging

## Abstract

Epidemiological studies show that having a history of cancer protects from the development of Alzheimer’s Disease (AD), and vice versa, AD protects from cancer. The mechanism of this mutual protection is unknown. We have reported that the peripheral blood mononuclear cells (PBMC) of amnestic cognitive impairment (aMCI) and Alzheimer’s Disease (AD) patients have increased susceptibility to oxidative cell death compared to control subjects, and from the opposite standpoint a cancer history is associated with increased resistance to oxidative stress cell death in PBMCs, even in those subjects who have cancer history and aMCI (Ca + aMCI). Cellular senescence is a regulator of susceptibility to cell death and has been related to the pathophysiology of AD and cancer. Recently, we showed that cellular senescence markers can be tracked in PBMCs of aMCI patients, so we here investigated whether these senescence markers are dependent on having a history of cancer. Senescence-associated βeta-galactosidase (SA-β-Gal) activity, G0-G1 phase cell-cycle arrest, p16 and p53 were analyzed by flow cytometry; phosphorylated H2A histone family member X (γH2AX) by immunofluorescence; IL-6 and IL-8 mRNA by qPCR; and plasmatic levels by ELISA. Senescence markers that were elevated in PBMCs of aMCI patients, such as SA-β-Gal, Go-G1 arrested cells, IL-6 and IL-8 mRNA expression, and IL-8 plasmatic levels, were decreased in PBMCs of Ca + aMCI patients to levels similar to those of controls or of cancer survivors without cognitive impairment, suggesting that cancer in the past leaves a fingerprint that can be peripherally traceable in PBMC samples. These results support the hypothesis that the senescence process might be involved in the inverse association between cancer and AD.

## 1. Introduction

Ample evidence from epidemiological studies has reported an inverse association between cancer and Alzheimer’s Disease (AD) in both directions, such that having a history of cancer protects from the development of AD and, conversely, suffering from AD is associated with lower odds of developing cancer [1,2,3,4,5,6,7,8,9,10,11,12]. This mutual protection has been reported for all cancer types, including skin cancers in most studies. Frain et al., 2011, [7] in a US cohort of 3,499,378 veterans found an inverse association for the majority of cancers, except prostate. Nonetheless, other reports have shown the inverse association with prostate cancer [13,14]. The cause of this mutual protection is unknown, but a biological mechanism deregulated in opposite directions has been proposed as an explanation [13,14,15,16]. Understanding the biological mechanism underlying the inverse association between cancer and AD is important since it might lead to new therapeutic targets for both diseases. In the study of the mechanism of the mutual protection, we previously reported that peripheral blood mononuclear cells (PBMCs) obtained from AD patients have a higher susceptibility to death when exposed to hydrogen peroxide (H_2_O_2_), and instead those from cancer survivors (Ca) are more resistant to oxidative death [17,18]. Interestingly, PBMCs from patients with both disorders, that is, with a history of cancer who developed cognitive impairment, showed a susceptibility to cell death that was more like that of Ca survivors than AD patients, suggesting that cancer might leave a trait that protects from AD [19]. In accordance with this hypothesis, we recently reported that patients with amnestic Mild Cognitive Impairment (aMCI) and a history of cancer in the past had a slower progression of cognitive decline, measured with the Montreal Cognitive Assessment (MoCA), compared with aMCI patients without cancer history [20].

Cellular senescence is a phenomenon in which cells arrest their growth and acquire resistance to cell death in response to several stress factors, such as aging, oxidative stress, DNA damage, and oncogenic stressors. Senescent cells characteristically secrete a series of inflammatory cytokines, growth factors, and proteases into the surrounding medium, called senescence associated secretory phenotype (SASP) [21,22]. Senescent cells accumulate in aging tissues and organs, thereby impairing physiological processes contributing to organismal aging [21,23,24]. Recently, cellular senescence has been implicated in the pathogenesis of AD and other neurodegenerative disorders [22,25,26,27]. Senescent cells have been reported in several brain cells [26,28,29,30,31,32] and, furthermore, the removal of senescent cells in animal models with senolytics, such as Quercetin and Dasatinib, is accompanied by reduced beta amyloid burden in the brain and an improvement in performance in learning and memory tests [26,33,34,35]. On the other hand, the importance of senescence in cancer has long been recognized. A protective role of senescence against cancer has been described by limiting cell proliferation. However, SASP release has effects on the surrounding tissues which could generate a cancer-promoting microenvironment [36,37,38].

We have recently reported that several senescence hallmarks can be tracked in PBMCs of patients suffering from aMCI and AD [27]. We found that PBMCs from aMCI patients have an increase in senescence-associated beta-galactosidase (SA-β-Gal) activity, an elevated percentage of cells arrested at G0-G1 and an increased expression of interleukins 6 (IL-6) and 8 (IL-8), whereas PBMCs from AD patients showed increased p16 and p53 expression and decreased γH2AX activity, compared with healthy controls (see panels A and B of scheme in Figure 1). We postulate that senescence could play a role in the mechanism of inverse association between AD and cancer and therefore, in the present study, we investigated whether a history of cancer could re-establish the markers of senescence identified in subjects with aMCI.

## 2. Results

### 2.1. Senescent Markers in PBMCs of aMCI Patients with and without a History of Cancer and in Cognitively Normal Cancer Survivors as Compared to Healthy Controls

As mentioned above, in a previous report we demonstrated that PBMCs of aMCI patients have increased levels of several senescence markers [27] (see panels A and B of scheme in Figure 1).

We here explored whether having a history of cancer imprints a change in these senescence markers in PBMCs of aMCI patients. We compared patients with aMCI and a history of any type of cancer (Ca + aMCI) to those of MCI patients without a history of cancer (aMCI), as well as with cognitively unimpaired cancer survivors (Ca survivors), relative to healthy control values. The values of senescence markers of aMCI patients and healthy controls were previously reported [27] and the comparisons were made relative to healthy controls (represented as a dotted line in all the figures).

The demographic and clinical characteristics of participants are shown in Table 1.

We first measured SA-β-Gal activity, one of the most accepted markers of cellular senescence, through the cytometric determination of the fluorescent product derived from the fluorogenic substrate C12FDG in PBMCs of patients from the different groups. As shown in Figure 2A, the increased level of SA-β-Gal activity present in PBMCs of aMCI patients [27] was not observed in PBMCs of Ca + aMCI patients, which showed levels similar to those of healthy controls (dotted line) and Ca survivors.

Similarly, the determination of PBMCs arrested at G0-G1, another key feature of senescence, showed that the Ca + aMCI group had percentages of G0-G1 that were comparable to those of healthy controls, and significantly lower than aMCI (Figure 2B). The Ca survivor group also showed mean Go-G1 percentages comparable to controls, but with high dispersion. The cause of this dispersion is not clear, but one possible explanation might be that the group included participants with different types of cancers (see Table 1).

p16 and p53 are well accepted markers of senescence as indicators of the p53–Cdkn1a (p21) and retinoblastoma (RB)–Cdkn2a (p16) pathways, respectively. Previously, we showed that both p16 and p53 were elevated in PBMCs of AD patients, without changes in the aMCI group [27]. Accordingly, we here found no changes in the determinations of p16 or p53 expression between aMCI, Ca + aMCI or Ca groups, all of which were non-significantly elevated compared to controls (Figure 3A,B).

Senescent cells activate the DNA damage response (DDR) that phosphorylates the orchestrator histone H2AX, which is used as a marker of senescence [39,40]. We determined the phosphorylated levels of H2AX (γH2AX) by immunofluorescence in PBMCs. In our previous study, we reported that PBMCs of aMCI patients had γH2AX levels that were not significantly lower than in healthy controls, which was not in accordance with the increase observed in other senescence markers. This might be explained since it has been suggested that the increase in γH2AX might be transient and there is a balance between the generation of DNA damage and the activation of repair pathways [40,41]. We here found that patients with a history of cancer had significantly lower γH2AX levels than aMCI (Figure 4G) and healthy controls (*p* < 0.005), suggesting that having had cancer leaves a trait in accordance with our hypothesis. As with other markers, the Ca + aMCI group had levels closer to the Ca survivor group than the aMCI group (Figure 4G).

### 2.2. Profile of Senescence-Associated Secretory Phenotype of aMCI Patients with and without a History of Cancer and in Cognitively Normal Cancer Survivors as Compared to Healthy Controls

Senescent cells characteristically secrete cytokines, immune modulators, growth factors and proteases and other molecules as part of the senescence-associated secretory phenotype known as SASP [42]. We determined the mRNA expression of cytokines IL-6 and IL-8 in PBMCs and their plasmatic levels. As with other senescent markers, we found that the Ca + aMCI group had IL-6 and IL-8 mRNA levels comparable to healthy controls and cancer survivors and lower than the aMCI group (Figure 5A,B). The levels in cancer survivors were significantly lower than in aMCI and Ca + aMCI groups (Figure 5A,B).

We also determined the plasmatic levels of IL-6 and IL-8 by ELISA. Ca + aMCI patients had significantly increased levels of IL-6 compared with aMCI (and healthy controls), although with a high dispersion of the values (Figure 5C). Cancer survivors had plasmatic IL-6 levels comparable to healthy controls. Interestingly, as seen with other senescence markers, the increased plasmatic levels of IL-8 seen in aMCI patients were significantly reduced in the Ca + aMCI group, attaining values like those of controls and cancer survivor groups, again favoring the suggestion that having had cancer leaves a trail.

## 3. Discussion

In this report, we demonstrate that patients who have a history of cancer before the development of aMCI do not show the hallmarks of senescence that we detected in PBMCs of aMCI patients, and instead show values like those of healthy controls and cancer survivors. The results support the suggestion that having suffered from cancer in the past leaves a trait that can be traceable in PBMC samples, and that senescence might be involved in the mechanism of the inverse association between AD and cancer. In Figure 1, we represent a scheme showing the changes observed in the present work (Panels C and D) compared with those described in the previous manuscript in aMCI and AD patients (Panels A and B) [27].

The results showing an increase in senescence hallmarks at the MCI phase of the disease might be interpreted by suggesting that senescence is deleterious for cognition, since it is not observed in the absence of cognitive impairment; that is, in healthy controls or cancer survivors. Alternatively, the increase in senescence markers in PBMCs in aMCI patients, which seems to be transitory since it was not seen at more advanced stages of AD [27], might represent an attempt to combat the process of neurodegeneration which later becomes exhausted. An increased senescence reflects a series of abnormalities and deregulations at the cellular level, such as lysosomal and autophagic abnormalities, dysregulated mTOR signaling, a decrease in mitochondrial membrane potential, enhanced ROS production and protein misfolding, among others (reviewed in [38]). This implies that aMCI patients have impaired cellular function in peripheral cells, reflecting a systemic effect of the pathology. However, patients undergoing this same neurodegenerative disease but with a history of previous cancer (Ca + aMCI) have fewer senescent markers and therefore preserved cell functioning that allows them to better respond to the neurodegenerative process. These results are in accordance with our report, showing a slower susceptibility to cell death induced by H_2_O_2_ [17,18] and with a slower progression of cognitive impairment than we reported in a subgroup of Ca + aMCI patients, compared with aMCI patients without a cancer history [20]. In this regard, it would be interesting to evaluate the proportion of senescence brain cells in this patient group.

PBMCs of Ca + aMCI patients showed fewer cellular senescence markers than aMCI patients measuring two key features: SA-β-Gal activity and the percentage of G0-G1 arrested cells. These markers were increased in aMCI; however, they were like those of healthy controls and Ca survivors in Ca + aMCI. We did not find differences among the groups in the determination of p16 or p53 mRNA expression, senescence markers that form part of the retinoblastoma (RB)–Cdkn2a (p16) and p53–Cdkn1a (p21) pathways, respectively, and which are activated during the cell-cycle blockade in senescent cells [43]. However, as previously shown, these markers were not elevated at the aMCI stage, but were elevated in AD patients [27].

We also measured γH2AX, an orchestrator histone that is phosphorylated and activated by the DNA damage response (DDR) during senescence and is therefore used as a senescence marker [39,40]. In the previous paper, we found that γH2AX activity was significantly reduced in PBMCs of AD patients and non-significantly reduced in the aMCI group. This reduction is counterintuitive since it would be expected to be increased similarly to other markers of senescence. An explanation for this discrepancy might be that γH2AX activation reflects that DNA damage response is a dynamic phenomenon, with a balance between the generation of molecular damage and the activation of repair pathways, which is compatible with reports showing a transient activation of γH2AX in response to cytotoxic insults [40,41]. On the other hand, there are controversial data in the literature on the activation of γH2AX during senescence, with some reports showing an increase and others a decrease in γH2AX activity in AD [39,44]. We found, here, that PBMCs of Ca survivor patients had significantly lower γH2AX activity than PBMCs from aMCI and healthy controls. This is interesting since it suggests that cancer in the past might have left this reduced activation. The levels of γH2AX activity were non-significantly reduced in the Ca + aMCI group compared with aMCI but were significantly lower than in healthy controls. Nevertheless, as with the other markers, the levels of this senescence marker in Ca + aMCI patients are more like those of the Ca group than the aMCI group, in support of the hypothesis of a protective effect of cancer. It must be taken into consideration that γH2AX activity was measured using whole PBMCs and there might be differences in γH2AX according to the various cell types of PBMCs. Nevertheless, our intention was to measure changes in the whole population of PBMCs to eventually use these measures to track the effect of treatments such as senolytics.

The measurement of SASP components such as mRNA expression of interleukins IL-6 and IL-8 by PBMCs also showed that the elevated levels seen in aMCI were not observed in patients who had Ca history and aMCI. The plasmatic levels of IL-6 were increased in the Ca + aMCI group, without a concomitant increase in the expression of IL-6 by PBMCs, indicating that other cells in blood or other organs must be contributing to plasmatic IL-6. We also observed a high dispersion of the values in this group. The origin of this high dispersion in Ca + aMCI patients is not clear but, as indicated above, it could be due to the different types of cancers of the participants in the group (see Table 1). As with other senescence markers, the plasmatic levels of IL-8 were lower in Ca + aMCI than in aMCI patients.

In all, we found that several senescence markers in peripheral cells, which we previously reported as elevated in aMCI patients [27], were decreased in patients that had both a history of a previous cancer and aMCI, with values more like those of controls or cancer survivors. These results were consistent with the hypothesis that cancer in the past leaves a fingerprint that protects from AD.

Beyond all the strategies used to assess senescence in this work, others have proposed studies of large genetic panels. Saul et al. [45] described a set of 125 genes that identify senescent cells and predict senescence-associated pathways in human samples. Along the same lines, specific DNA methylation patterns that regulate the expression of relevant genes for the development of the senescence phenotype have also been evaluated [46,47].

The role of senescence in aging and several systemic diseases, and more recently in the brain and neurodegenerative diseases, is increasingly being recognized as important. Furthermore, the development of senolytics to remove senescent cells, which was shown to improve cognitive ability in mouse models, makes the study of senescence very necessary. The search for peripheral markers of cellular senescence is a matter of great interest in geroscience, since it will allow the estimation of the accumulation of senescent cells in patients, as well as the controlling of the effect of an intervention with senolytics. In this search for clinical applications, our study has evaluated senescence markers in PBMC and not in lymphocyte subpopulations. Although there might be differences between these subgroups, we believe that the measurement of senescent markers in PBMCs is a simpler method to measure in clinical laboratories.

## 4. Materials and Methods

### 4.1. Subjects

Participants were recruited from a longitudinal database of one of the authors (MIB) at the Centro Investigación Clínica Avanzada, Hospital Universidad de Chile (CICA-HCUCH), where patients had been followed for 15 years with evaluations every six or twelve months. Patients with a history of cancer who later developed cognitive impairment (Ca + aMCI) and those with a history of cancer without cognitive impairment were recruited from the Hospital Clínico Universidad de Chile. Patients with aMCI and healthy controls were those from a previous publication. The study was approved by the Ethics Committee of the Hospital Clínico Universidad de Chile, and all participants signed an informed consent form. A diagnosis of aMCI was performed using the NIA-AA criteria [48]. Cognitive impairment severity was assessed using the CDR and CDR-SOB [49], the Montreal Cognitive Assessment (MoCA) [50] validated for the Chilean population [51] and the MoCA-Memory Index Score (MoCA-MIS) [52]. An informant filled in the Alzheimer’s Disease 8 (AD8) Questionnaire [53]. A history of cancer was obtained from clinical charts and remote anamnesis with the patient and their informants. All cancer types were recruited (including skin cancer, basal cell carcinoma (BCC), squamous cell carcinoma (SCC) and melanoma). Baseline socio demographic data (age, sex, years of education), baseline comorbidities (hypothyroidism, type 2 diabetes [DM2], high blood pressure [HBP], hypercholesterolemia) and treatment status (memantine and/or acetylcholinesterase [AChE] inhibitors) were also obtained from clinical records. The demographic characteristics of the study participants are shown in Table 1.

### 4.2. Peripheral Blood Mononuclear Cells (PBMCs) Isolation

Peripheral blood was collected by venipuncture in sodium heparin vacutainer tubes following a standardized procedure; all blood samples were obtained between 10 and 12 am (non-fasting) and processed within 2 h after venipuncture. PBMCs were separated from whole blood by Ficoll-Hypaque TM PLUS (Merk. Darmstadt, Germany), density centrifugation, as previously described [18,54]. The viability of PBMCs was routinely checked and only those experiments in which cell survival was higher than 95% were included. The proportion of different PBMC cells was evaluated by flow cytometry, and in general the distribution was 16.5% of monocytes, 8.5% of B cells and 58% of T cells, without major differences between groups. Cells were preserved in 1 mL TriZol Reagent (ThermoFisher, Waltham, MA, USA) and stored at −80 °C until mRNA extraction. For β-Galactosidase activity, GoG1 arrest and immunofluorescence, fresh PBMCs were used.

### 4.3. β-Galactosidase Activity

The β-galactosidase activity was measured by a fluorescence method [55] as described in [27]. Briefly, PBMCs were incubated with 100 nM bafilomycin A1 (Merk. Darmstadt, Germany) in RMPI-1640 medium (ThermoFisher, Waltham, MA, USA) for 1 h, followed by the addition of 33 µM 5-dodecanoylaminofluorescein di-β–d-galactopyranoside (C_12_FDG) (ThermoFisher, Waltham, MA, USA) and incubation for 2 more hours in the same conditions. The cells were washed 3 times with ice-cold phosphate-buffered saline (PBS) and re-suspended in PBS for the flow cytometry determination of mean fluorescence intensity (MFI) to estimate β-galactosidase activity.

### 4.4. p16 and p53 Determination

p16 and p53 expressions were determined by flow cytometry as described in [27]. Briefly, cryopreserved PBMCs were resuspended in a PBS solution supplemented with 10% fetal bovine serum, washed with PBS 3% fetal bovine serum (FBS) and fixed using the Cytofix-CytopermTM kit (BD Biosciences, Franklin Lakes, NJ, USA). Fixed cells were incubated with anti-human p16 (BD Pharmingen, Franklin Lakes, NJ, USA) and anti-human p53 (BD Pharmingen, Franklin Lakes, NJ, USA) antibodies for 16 h at 4 °C. The secondary antibody was Alexa 488 anti-rabbit (Thermo Fisher Scientific, Waltham, MA, USA) 1:1000 in PBS 3% FBS for 1 h at room temperature. Quantification was performed using a Cytoflex (BD Biosciences, Franklin Lakes, NJ, USA) flow cytometer and FlowJo software v7.6.1, considering a cell population of 100,000 cells for each analysis.

### 4.5. G0G1 Phase Cell-Cycle Arrest

The percentage of PBMCs arrested at the G0G1 phase in the cell cycle was determined by flow cytometry with propidium iodide. Briefly, PBMCs stained with propidium iodide were analyzed by flow cytometry (FACSCanto BD Biosciences, Franklin Lakes, NJ, USA) and cell count versus linear fluorescence was used to create a histogram of the distribution of DNA content across the stages of the cell cycle. The proportion of cells in the G0/G1 cycle were compared in the different groups.

### 4.6. H2AX Immunocytochemistry

PBMCs were fixed (1% formaldehyde fixation solution PBS for 10 min) and rinsed three times with PBS, followed by incubation for 5 min in 0.1% Triton X-100 (Merck, Darmstadt, Germany) in PBS and then blocked with 2% BSA in PBS for 30 min. Immunostaining of PBMCs was performed with anti-γH2AX (ser139) as the primary antibody (Merck, Darmstadt, Germany) (1:3000, diluted in 0.2% BSA in PBS) at 4 °C overnight, then rinsed three times with PBS and incubated with Alexa Fluor^®^ 488 (ThermoFisher, Waltham, MA, USA) anti-mouse as the secondary antibody (1:400, diluted in 0.2% BSA in PBS) for 1 h at room temperature. Cells were rinsed three times with PBS, and the coverslips were mounted in DAKO mounting medium (Abcam, Cambridge, UK) on glass slides.

Confocal image stacks were captured with a 20× objective in a Nikon C2+ microscope (Tokyo, Japan). Around 100 to 120 cells acquired per patient were used to generate the Zeta projections from 10 to 15 stacks (0.8 mm thickness each) generated with ImageJ 1.53t analysis software. Immunofluorescence data were quantified with histogram analysis of the fluorescence intensity at each pixel across the nucleus and normalized by the number of nuclei and expressed in arbitrary fluorescence units (AU).

### 4.7. RNA Isolation and PCR Analysis

Total RNA was isolated using Trizol reagent (ThermoFisher, Waltham, MA, USA). A DNAse digestion step with TURBO DNA-freeTM Kit was included to remove any contaminating genomic DNA. cDNA was synthesized from total 2 µg RNA using the High-Capacity cDNA Reverse Transcription Kit. Real-time quantitative PCR (qPCR) was performed in an AriaMx real-time PCR System (Agilent Technologies) using the DNA binding dye SYBR green (Brilliant II SYBER-GREEN Master Mix). Amplification was performed using the following primers: IL6: forward 5′-AACTCCTTCTCCACAAGCGCC-3′, reverse 5′-GTGGGGCGGCTACATCTTT-3′; IL8: forward, 5′-CTCTCTTGGCAGCCTTCCTGATT-3′, reverse 5′-AACTTCTCCACAACCCTCTGCAC-3; 18S: forward 5′-GATATGCTCATGTGGTGTTG-3′, reverse 5′-AATCTTCAGTCGCTCCCA-3′; SDHA: forward 5′-GAGGCAGGGTTTAATACAGCA-3′, reverse 5′-CCAGTTGTCCTCCTCCATGT-3′. All genes were normalized to the geometric mean of 18S and SDHA housekeeping genes. All samples were run in triplicate.

### 4.8. ELISA

The plasma levels of IL-6 and IL-8 were determined by ELISA, following the manufacturer’s recommendations. The Human IL-6 High Sensitivity ELISA Kit from eBioscience (Waltham, MA, USA) and the Human Ultrasensitive ELISA Kit from Thermo-Fisher (Waltham, MA, USA) for IL-8 were used.

### 4.9. Statistical Analysis

Results between the groups were analyzed using Kruskal–Wallis with Dunn’s post hoc test for multiple comparisons. The data in each group were corrected for age and sex using multiple regression. Percentages were compared with proportion statistics. All statistical analyses were performed using the GraphPad Prism 9 software version 9.1.1. All values were expressed as the mean ± SEM. A *p*-value < 0.05 was considered statistically significant for all measurements.

## 5. Conclusions

We compared the presence of senescence markers in PBMCs of aMCI patients with those of patients with both aMCI and a history of cancer. We found that patients with both cancer and aMCI did not show the increase in several senescence markers that was observed in aMCI patients without cancer history, showing values within the range of healthy controls or cancer survivors without cognitive impairment, supporting the hypothesis that cancer might leave a footprint in the senescence process that grants a protective effect against AD. These results support the hypothesis that the senescence process might be involved in the inverse association between cancer and AD. Additionally, it is interesting to highlight the possibility of measuring senescence markers in peripheral cells, which offers a simple opportunity to track the effect of treatments such as senolytics.

## Figures and Tables

**Figure 1 ijms-24-07364-f001:**
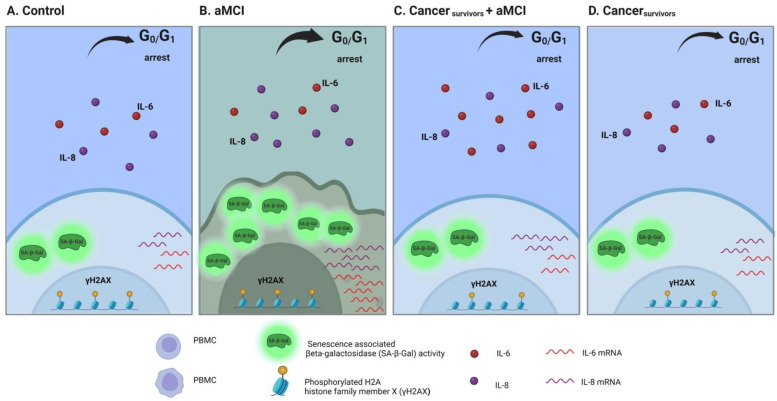
Schematic representation showing that a history of cancer avoids the increase of senescence markers in PBMCs of aMCI patients. The figures represent PBMCs of healthy controls (**A**), aMCI (**B**), Ca + aMCI (**C**), and Ca survivor (**D**) patients. Panels A and B represent the results reported in [27]. Senescence markers SA-β-Gal, Go-G1 arrested cells, IL-6 and IL-8 mRNA expression, and IL-8 plasmatic levels, which were elevated in PBMCs of aMCI patients, were decreased in those of Ca + aMCI to levels as those of controls or of cognitively normal cancer survivors.

**Figure 2 ijms-24-07364-f002:**
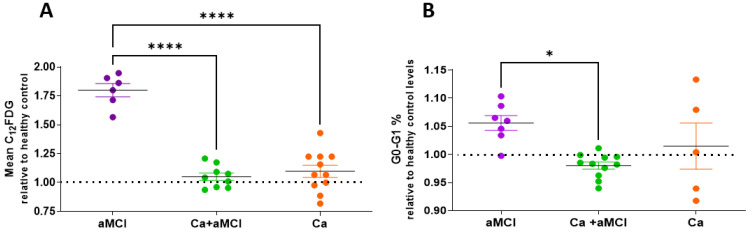
Senescent SA-β-Gal marker and growth arrest in PBMCs Ca + aMCI patients were lower than in aMCI patients and like those of cognitively normal Ca survivors and healthy controls. Values are mean C12FDG Units relative to healthy controls (dotted line, 18.5 ± 1.1 C12FDG Units (mean ± SE)). (**A**) SA-β-Gal activity in PBMCs of patients with Ca + aMCI (n = 9) was significantly decreased as compared with aMCI (n = 6), while cognitively normal cancer survivors (Ca, n = 11) had levels comparable to the Ca + aMCI group, both not different from healthy controls. (**B**) Growth arrest of PBMCs was determined by the percentage of cells at cell-cycle stage G0-G1 relative to the values of healthy controls (dotted line, 81.5 ± 1.3% (mean ± SE)). The percentage of PBMCs arrested at G0-G1 in Ca + aMCI (n = 11) patients was lower than in aMCI (n = 7) patients and comparable to Ca survivors (n = 5) and healthy controls. Statistical analysis: Kruskal–Wallis with Dunn’s post hoc correction, * *p* < 0.05; **** *p* < 0.0001.

**Figure 3 ijms-24-07364-f003:**
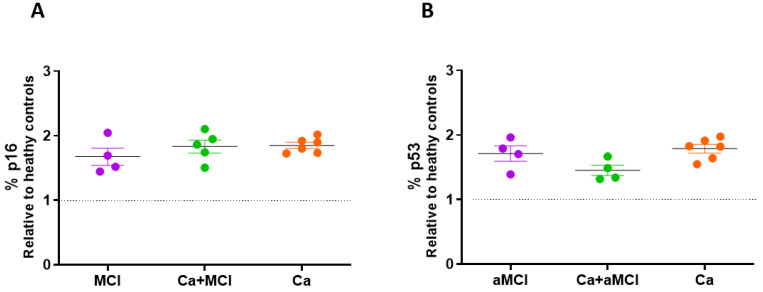
The expression of p16 and p53 was not significantly different in PBMCs of aMCI, Ca + aMCI and Ca survivors. Expressions of p16 (**A**) and p53 (**B**) were determined by flow cytometry in PBMCs from aMCI (n = 4), Ca + aMCI (n = 5) and Ca survivors (n = 6). The results are expressed relative to the mean value of healthy controls (dotted line, 40.2 ± 8.7% for p16; 39.0 ± 7.2% for p53, mean ± SE). Statistical analysis: Kruskal–Wallis with Dunn’s post hoc correction.

**Figure 4 ijms-24-07364-f004:**
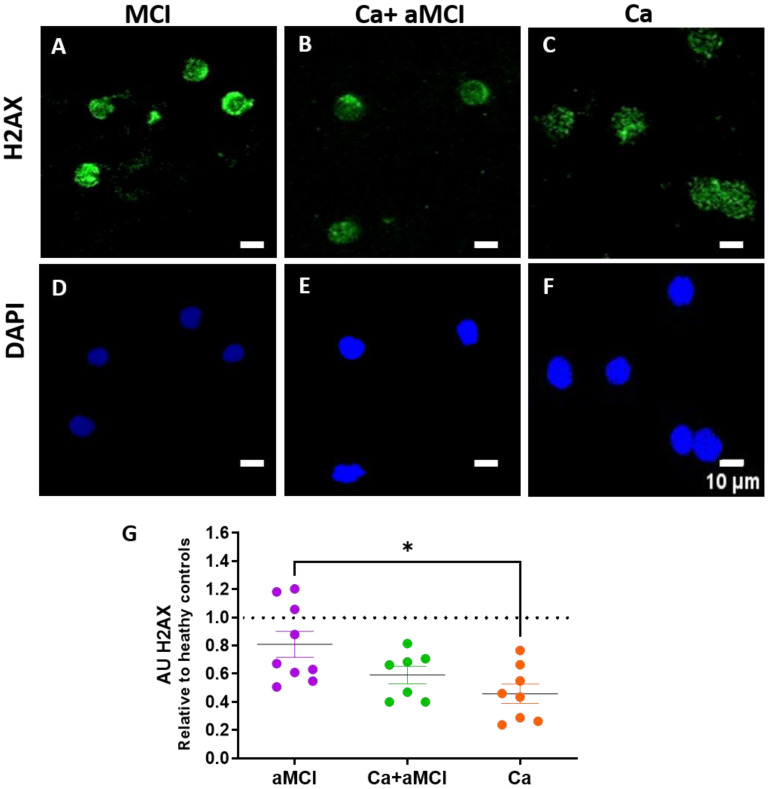
Activation of the DNA damage response, determined by immunofluorescence γH2AX activity in PBMCs of aMCI, Ca + aMCI and Ca survivors. Representative immunofluorescence confocal images of γH2AX staining (green, (**A**–**C**)) and nuclei stained with DAPI (blue, (**D**–**F**)) in PBMCs of representative aMCI (**A**,**D**), aMCI + Ca (**B**,**E**) and Ca survivor (**C**,**F**) patients. Scale bars: 10 μm. (**G**): Immunofluorescence quantification of γH2AX activity normalized by the number of nuclei in aMCI (n = 9), aMCI + Ca (n = 7) and Ca survivor (n = 8) patients. γH2AX levels were decreased in PBMCs of Ca survivors compared with aMCI patients and healthy controls (dotted line; *p* < 0.05. Healthy controls had 12330 ± 608, mean ± SE, fluorescent arbitrary Units); those of Ca + aMCI patients were also decreased, although non-significantly. Around 98 to 100% of analyzed nuclei were positive for γH2AX without major differences between the different groups. Statistical analysis: Kruskal–Wallis with Dunn’s post hoc correction, * *p* < 0.05.

**Figure 5 ijms-24-07364-f005:**
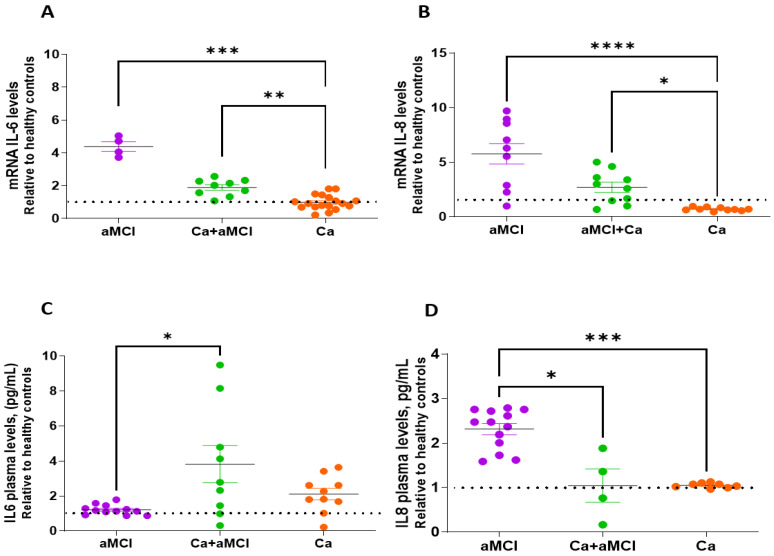
mRNA expression by PBMCs and plasmatic levels of IL-6 and IL-8 in patients with MCI, Ca + aMCI, and cancer survivors, relative to healthy control levels. (**A**,**B**): mRNA expression levels of cytokines IL-6 (**A**) and IL-8 (**B**) determined by qPCR in PBMCs of aMCI (n = 5), Ca + aMCI (n = 9), and Ca survivor (n = 18) patients expressed relative to healthy controls (dotted line, 565.6 ± 65.5 and 2.8 ± 0.3 Units, mean ± SE, for IL-6 and IL-8, respectively). The increased mRNA expressions of cytokines IL-6 and IL-8 by PBMCs of aMCI patients were not present in Ca + aMCI patients, nor in Ca survivors. PBMCs of Ca survivors had significantly lower mRNA expressions of both cytokines compared with aMCI and Ca + aMCI. (**C**,**D**): Plasmatic levels of IL-6 (**C**) and IL-8 (**D**) relative to healthy control values (dotted line, 0.54 ± 0.01 and 1.23 ± 0.02 pg/mL, mean ± SE, for IL-6 and IL-8, respectively) were measured by ELISA in PBMCs of aMCI (n = 12), Ca + aMCI (n = 9 for IL-6 and n = 4 for IL-8) and Ca survivors (n = 10 for IL-6 and n = 8 for IL-8). Statistical analysis: Kruskal–Wallis with Dunn’s post hoc correction, * *p* < 0.05, ** *p* < 0.01, *** *p* < 0.005, **** *p* < 0.001.

**Table 1 ijms-24-07364-t001:** Demographic data of participants.

	Controls * n = 26	aMCI * n = 22	Ca + aMCI n = 21	Ca Survivors n = 18	*p* Value
Age, mean ± SE (range)	75.6 ± 1.8 (65–85)	77.2 ± 1.5 (63–93)	79.5 ± 1.6 (69–90)	75.8 ± 1.9 (61–91)	ns
Female sex, n (%)	17 (68.0)	16 (72.7)	10 (47.6)	13 (72.2)	ns
Education, years	12.5 ± 1.0	10.1 ± 1.1	11.3 ± 1.8	14.3 ± 1.1	ns
MoCA test score, mean ± SE	28.2 ± 0.4	19.8 ± 0.9	22.5 ± 0.7	28.8 ± 0.2	*p* < 0.0001 for control vs. aMCI; control vs. Ca + aMCI; aMCI vs. Ca; and Ca + aMCI vs. Ca.
MoCA-MIS test score, mean ± SE	14.1 ± 0.2	8.7 ± 0.5	8.4 ± 0.9	14.3 ± 1.1	*p* < 0.0001 for control vs. aMCI; control vs. Ca + aMCI; aMCI vs. Ca; and Ca + aMCI vs. Ca
AD8, mean ± SE	0.4 ± 0.1	4.3 ± 0.5	3.6 ± 0.5	0.1 ± 0.08	*p* < 0.0001 for control vs. aMCI; control vs. Ca + aMCI; aMCI vs. Ca; and Ca + aMCI vs. Ca.
CDR-SOB, mean ± SE	0.1 ± 0.04	2.1 ± 0.2	2.6 ± 0.3	0.1 ± 0.04	*p* < 0.0001 for control vs. aMCI; control vs. Ca + aMCI; aMCI vs. Ca; and Ca + aMCI vs. Ca. *p* < 0.05 for aMCI vs. Ca + aMCI
CDR 0, n	26	0	0	18	
CDR 0.5, n	0	22	13	0	
CDR 1, n	0	0	8	0	
CDR 2, n	0	0	0	0	
CDR 3, n	0	0	0	0	
Diabetes/Insulin Resistance, n (%)	6 (22.2)	6 (25.0)	4 (19.1)	7 (38.0)	ns
Hypertension, n (%)	19 (72.2)	10 (45.0)	12 (57.1)	8 (44.4)	ns
Hypercholesterolemia, n (%)	7 (27.8)	10 (47.4)	10 (47.6)	3 (16.6)	*p* < 0.05: Ca + aMCI vs. Ca
Anti-Dementia medication					
AChE inhibitor + Memantine, n (%)	-	9 (40.9)	9 (42.9)	-	ns
AChE inhibitor only, n (%)	-	0	2 (9.5)	-	ns
Memantine only, n (%)	-	5 (22.7)	2 (9.5)	-	ns
Cancer type, n					
Skin			9 ^#^	14 ^$^	
Breast			5	5	
Prostate			2	3	
Colon			4	-	
Ovary			1	1	
Pancreas			-	1	
Cervix uterus			-	1	
Kidney			-	1	
Thyroid			-	1	
Gastric			-	1	
Bladder			1	1	
Lymphoma			1	-	
Head and neck			1	-	
Lung			1	-	
Rectum			1	-	
Endometrium			1	-	
Appendix				1	
Total Cancer (n)	0	0	27	30	

* Values for controls and aMCI participants are from [27]. Several patients had more than one cancer. ^#^ Nine basal cell carcinomas (BCC), (one BCC + melanoma). ^$^ Eleven BCC (one BCC + squamous cell cancer (SCC) + melanoma, one BCC + melanoma), two melanoma, one SCC. MoCA, Montreal Cognitive Assessment. AD8, Informant questionnaire Alzheimer’s Disease 8. CDR, Clinical dementia Rating. CDR-SOB, Clinical dementia Rating–Sum of Boxes. ns, non-significant. Statistical analysis: One-way ANOVA, Kruskal–Wallis test for mean comparisons and proportion statistics for sex and comorbidities comparisons.

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
