# Peer review of "Cancer History Avoids the Increase of Senescence Markers in Peripheral Cells of Amnestic Mild Cognitive Impaired Patients"

_ijms, 2023, doi:10.3390/ijms24087364_

Round 1

Reviewer 1 Report (Previous Reviewer 1)

Comments

1. Any specific reason to include only Female subjects in the current study? Does sex difference have a correlation with cancer or AD?

Author Response

We than the reviewer for revising our work. We would like to clarify that the patients recruited in this study were predominantly females (between 47% and 72%), but not only females, as shown in Table 1.

There are some indications that AD affects females more than men, but it is not clear whether this is really a higher incidence in females or only more prevalence because women live longer than men. Cancer on the other hand shows sex differences depending on the type of cancer, i.e. breast is more prevalent in women, prostate is only present in males etc.

Reviewer 2 Report (New Reviewer)

In this manuscript Carol D SanMartín, et al. described results of their work aimed at analysis of mechanisms of possible influence of cancer on development of neurodegenerative diseases, and vice versa, the anticancer impact of the neurodegenerative diseases. The inverse association of some neurodegenerative diseases with different types of human cancers was previously demonstrated in a number of publications, including the works of this group. This area of research is very relevant, because the study of the mechanisms of antitumor reactions of the body, and the mechanisms of processes that prevent the development of human neurodegenerative diseases is an urgent biomedical problem.

For the study the authors used peripheral blood mononuclear cells of four adequate groups of patients and healthy donors; application of modern biological methods was used for the investigation. The results of the study confirmed the earlier assumptions of the authors that one of the mechanisms of the relationship between tumor and neurodegenerative processes in the human body is associated with a pool of senescent cells in the peripheral blood mononuclear cells population. This may be the basis for the use of biomarkers of such cells and drugs for their elimination in clinical practice.

In general, the study is well presented, proper controls are used and the conclusions are convincingly supported by experimental results, the data are of considerable novelty and interest. Manuscript is well written, I have no suggestion to improve the quality of the manuscript.

The manuscript is suitable for publication.

Author Response

We sincerely thank the reviewer for his comments.

Reviewer 3 Report (New Reviewer)

1) As mentioned above, in a previous report we demonstrated that PBMCs of aMCI patients have increased levels of several senescence markers (see Fig 5). The results start with a strange reference to figure 5. And this is the previous data of the authors ... Should this figure be in the introduction or discussion?

2) 4.5. G0G1 Phase Cell-Cycle Arrest. This section should be described in great detail.

3) The conclusion is of course interesting, but it should be rewritten and reflect the results obtained by the authors. At a minimum, the conclusion should be expanded.

Author Response

We thank the reviewer for his comments and suggestions. 

1) As mentioned above, in a previous report we demonstrated that PBMCs of aMCI patients have increased levels of several senescence markers (see Fig 5). The results start with a strange reference to figure 5. And this is the previous data of the authors ... Should this figure be in the introduction or discussion?

We understand the comment expressed by the reviewer since it is confusing to start the Results section with Fig 5. To avoid this confusion, we added the reference of the previous work and the phrase “see scheme in Fig 5”. In addition, to further clarify this point we added a sentence in the legend to Fig 5 explaining that the panels A and B of the figure describe the results of our previous paper.

2) 4.5. G0G1 Phase Cell-Cycle Arrest. This section should be described in great detail.

We acknowledge that the description of the method of GoG1 Phase Cell-Cycle arrest detection was insufficient. We added a more complete description in the method section: “Briefly, PBMCs stained with propidium iodide were analyzed by flow cytometry (FACSCanto BD) and cell count versus linear fluorescence was used to create a histogram of the distribution of DNA content across the stages of the cell cycle. The proportion of cells in G0/G1 cycle were compared in the different groups.” 

3) The conclusion is of course interesting, but it should be rewritten and reflect the results obtained by the authors. At a minimum, the conclusion should be expanded.

Following the suggestion, we rewrote the Conclusions of the manuscript to better reflect the results of our work with the sentence: “We compared the presence of senescence markers in PBMCs of aMCI patients with those of patients with both aMCI and history of cancer. We found that patients with both cancer and aMCI do not show the increase in several senescence markers that were observed in aMCI patients without cancer history, showing values within the range of healthy controls or cancer survivors without cognitive impairment, supporting the hypothesis that cancer might leave a footprint in the senescence process that grants a protective effect against AD. These results support the hypothesis that the senescence process might be involved in the inverse association between cancer and AD. Additionally, it is interesting to highlight the possibility of measuring senescence markers in peripheral cells, which offers a simple opportunity to track the effect of treatments such as senolytics.”

This manuscript is a resubmission of an earlier submission. The following is a list of the peer review reports and author responses from that submission.

Round 1

Reviewer 1 Report

The manuscript entitled “Cancer history imprints resistance to cellular senescence in peripheral blood mononuclear cells of amnestic Mild Cognitive 3 Impaired patientsby Felipe Salech et. al. describes the measurement of cellular senescence markers in PBMCs of aMCI patients with a history of cancer.

Similar methodology is already published in Int J Mol Sci. 2022 Aug 20;23(16):9387 from the same group titled “Senescence Markers in Peripheral Blood Mononuclear Cells in Amnestic Mild Cognitive Impairment and Alzheimer’s Disease”. The current work is utilized the data from previous work with inclusion of data from aMCI patient with a history of cancer/ cancer survivors in the measurement of cellular senescence markers in PBMC’s to study the inverse relationship between AD and cancer.

Comments:

1.      It is well known that multiple physiological factors/ donor physiological status influences the PBMC. Although, use of increased donor increases the experimental variation, where recording PBMC over certain period (ex :2-5 years) from the same donor will support the generality of the result. Missing longitudinal study in the current work

2.      Given the PBMC are a heterogenous population of cells what is the quality, and distribution of cells in PBMC in the current method?

3.      Time of blood sample collection morning or evening time? also, the processing time of PBMCs may also be an important variable. Authors should present this in the methods section.

4.      Authors describes that PBMC samples from several donor with Ca+aMCI/Ca leaves a senescence trait. Does any specific cancer alter this inverse relationship? Is there any study reported?

5.      Senescence-associated gene map for cells population (ex: - memory and naïve T-cell) from PBMC of Ca+aMCI with omics data would be helpful to support the current study.

6.      All the figures in the manuscript should be presented with clear statistical significance. The flagging with stars in all the figures requires correction.

7.      Figure 1b: Why Ca, G0-G1% data has high variance? This raises the concern on the physiological status of PBMC sample from cancer survivors. Also, Figure 6c.

8.      What is the Epigenetic status (DNA methylation status) of the PBMC, from all the groups of patients? Which need to be discussed in the current study.

9.      Figure 3. Should be represented as quantification of γ-H2AX foci /nuclei for each individual rather than γ-H2AX expression.

10.  Dependency/Differential mRNA expression of IL6- and IL8 with plasma levels of protein abundance was not clear.  IL6 and IL8 plasma levels should be correlated with mRNA expression levels?  I could see an mRNA-protein correlation with IL8 but not with IL-6.

11.  What is the status of H2AX in the groups with or without DNA damaging agent? Do they still see the reduced expression of H2AX in aMCI, Ca+aMCI and Ca?   patients are in any medication while collecting the samples

Author Response

We thank the reviewer for their comments and questions which we answered point by point. We marked the changes using Track changes as indicated. The references were corrected (since we added a few).

“Cancer history imprints resistance to cellular senescence in peripheral blood mononuclear cells of amnestic Mild Cognitive 3 Impaired patients” by Felipe Salech et. al. describes the measurement of cellular senescence markers in PBMCs of aMCI patients with a history of cancer.

Similar methodology is already published in Int J Mol Sci. 2022 Aug 20;23(16):9387 from the same group titled “Senescence Markers in Peripheral Blood Mononuclear Cells in Amnestic Mild Cognitive Impairment and Alzheimer’s Disease”. The current work is utilized the data from previous work with inclusion of data from aMCI patient with a history of cancer/ cancer survivors in the measurement of cellular senescence markers in PBMC’s to study the inverse relationship between AD and cancer."

Comments:

  1. It is well known that multiple physiological factors/ donor physiological status influences the PBMC. Although, use of increased donor increases the experimental variation, where recording PBMC over certain period (ex :2-5 years) from the same donor will support the generality of the result. Missing longitudinal study in the current work.

We agree with the reviewer 's comment about the multiple physiological factors and status of PBMCs of the different participants. We recruited volunteers from a longitudinal database at the Centro Investigación Clínica Avanzada, Hospital Universidad de Chile (CICA-HCUCH where participants have been followed for 15 years with evaluations every 6 months or at least annually. Therefore, the diagnosis of the included patients has been corroborated in several subsequent evaluations. We have also made repeated measures of senescence and cell death in PBMCs at different time points in the same patient with similar results. The viability of the PBMCs after every extraction is routinely checked and only those experiments in which cell survival is higher than 95 % are included. The data in this study corresponds to single measures of senescence for each patient. We added a paragraph in Methods indicating this.

  1. Given the PBMC are a heterogenous population of cells what is the quality, and distribution of cells in PBMC in the current method?

As mentioned above the viability of PBMCs after extraction is routinely checked and only those samples in which the viability is over 95% are included in the study. The proportion of different PBMCs cells was checked by flow cytometry. In general, the distribution is 16.5% of monocytes, 8.5% of B cells and 58% of T cells without major differences between the different groups. We added a sentence in methods stating this point.

  1. Time of blood sample collection morning or evening time? also, the processing time of PBMCs may also be an important variable. Authors should present this in the methods section.

We thank the reviewer for pointing out this issue and we added a paragraph in methods indicating the standardized procedure that we use; all blood samples are taken between 10 and 12 am (nonfasting) and processed within 2 hours after venipuncture. 

  1. Authors describes that PBMC samples from several donor with Ca+aMCI/Ca leaves a senescence trait. Does any specific cancer alter this inverse relationship? Is there any study reported?

We acknowlege this question, and we have added the following paragraph in the introduction to answer it: “It is not clearly established whether the inverse association is for specific cancers. Most studies have found the inverse association analyzing all types of cancers, including skin cancers. Few reports have enough cases to investigate whether specific cancers are involved in the process of mutual protection. Frain et al 2011 [7] in a US cohort of 3,499,378 veterans found the inverse association for the majority of cancers, except prostate. Nonetheless, other reports have shown the inverse association with prostate cancer [14,15]. (Ibañez et al 2014, Sanchez-Valle 2017)”. Therefore, the general belief is that cancer in general is inversely associated with AD.

  1. Senescence-associated gene map for cells population (ex: - memory and naïve T-cell) from PBMC of Ca+aMCI with omics data would be helpful to support the current study.

We thank the suggestion, and we agree that a study of large genetic panels would be very interesting to support our results. We added a paragraph in the discussion sections to address this issue. “Beyond the strategies we used to assess senescence in this work, others have proposed studies of large genetic panels. Dominik et al reported on a set of 125 genes to identify senescent cells and predict senescence-associated pathways through tissues in human samples (Saul et al Nature Communications 2022). Along the same line, specific DNA methylation patterns that regulate the expression of relevant genes for the development of the senescence phenotype have also been evaluated (Xie Cancer Cell 2018, Pepin, Front Genet 2020).

  1. All the figures in the manuscript should be presented with clear statistical significance. The flagging with stars in all the figures requires correction.

We acknowledge that there was a bad transcription of the asterisks in the figures in the pdf version of the manuscript. We corrected this error in all the figures incorporating TIFF figures.

  1. Figure 1b: Why Ca, G0-G1% data has high variance? This raises the concern on the physiological status of PBMC sample from cancer survivors. Also, Figure 6c.

We agree with the reviewer that there was high variability of the % of cells arrested at G0/G1 in cancer survivors (Fig 1 B) and of the plasmatic levels of IL-6 in Fig 4 C in the Ca+aMCI group. As explained above the survival of PBMCs was checked, so we believe this is not due to variability of the physiological status of PBMCs but rather to other factors, such as the different types of cancer of the patients in each group. Furthermore, if it were due to the physiological status of PBMCs, a similar variability would be expected in all the groups and not only in some of them. We added a sentence in the results section (pages 6 and 9).

  1. What is the Epigenetic status (DNA methylation status) of the PBMC, from all the groups of patients? Which need to be discussed in the current study.

We agree with the reviewer on the relevance of the epigenetic status of the PBMCs of the participants in our study. We added a paragraph in the Discussion addressing this issue: “Beyond all the strategies used to assess senescence in this work, others have proposed studies of large genetic panels. Dominik et al described a set of 125 genes that identify senescent cells and predict senescence-associated pathways in human samples (Dominik Nature Communications 2022). Along the same line, specific DNA methylation patterns that regulate the expression of relevant genes for the development of the senescence phenotype have also been evaluated (Xie Cancer Cell 2018, Pepin, Front Genet 2020.

  1. Figure 3. Should be represented as quantification of γ-H2AX foci /nuclei for each individual rather than γ-H2AX expression.

We represented the total intensity of γ-H2AX divided by the number of nuclei. We quantified that around 98 to 100% of nuclei were stained for γ-H2AX, without major differences between the groups. We added a sentence in the legend to figure 3 to clarify this point as suggested.

  1. Dependency/Differential mRNA expression of IL6- and IL8 with plasma levels of protein abundance was not clear. IL6 and IL8 plasma levels should be correlated with mRNA expression levels?  I could see an mRNA-protein correlation with IL8 but not with IL-6.

As the reviewer mentions we found a correlation between mRNA expression and plasmatic levels of IL-8, but this was not seen with IL-6, where we found that the Ca+aMCI group had higher IL-6 plasmatic levels (compared with controls) without a concomitant increase in mRNA expression by PBMCs. This result suggests that other sources (cells in the blood or elsewhere) must be contributing with plasmatic IL-6. We changed the wording in results and discussion to better explain this idea.

  1. What is the status of H2AX in the groups with or without DNA damaging agent? Do they still see the reduced expression of H2AX in aMCI, Ca+aMCI and Ca? patients are in any medication while collecting the samples

The status of H2AX after a damaging agent is an interesting issue to study. However, in this study we did not evaluate the effect of a damaging agent; we report the activity of H2AX in basal conditions.

We added in Table 1 the anti-dementia medications that the patients were taking. There was no significant difference in the anti-dementia medications between the aMCI and Ca+aMCI groups.

Reviewer 2 Report

General comments:

This manuscript tests the hypothesis that cancer might leave a trait in the senescence process that protects from the develop-42 ment of cognitive deterioration.

Minor comments:

1. line 35-36: “ Senescence-asso-35 ciated βeta-galactosidase (SA-β-Gal) activity, G0-G1 phase cell-cycle arrest, p16 and p53 were ana-36 lyzed by flow cytometry”. Were p16 and p53 detected by flow cytometry?

2. line 166: “mRNA expression of p16 (A) and p53 (B) were determined by flow cytometry”. Is the flow cytometry correct?

3. How p21 and p53 were detected? I cannot find it in Section Materials and methods. Please provide a detailed information.

Author Response

We thank the reviewers for their comments and questions which we answered point by point. We marked the changes using Track changes as indicated. The references were corrected (since we added a few).

"Comments and Suggestions for Authors

General comments:

This manuscript tests the hypothesis that cancer might leave a trait in the senescence process that protects from the develop-42 ment of cognitive deterioration.

Minor comments:

  1. line 35-36: “ Senescence-asso-35 ciated βeta-galactosidase (SA-β-Gal) activity, G0-G1 phase cell-cycle arrest, p16 and p53 were ana-36 lyzed by flow cytometry”. Were p16 and p53 detected by flow cytometry?"

We thank the reviewer for noticing that we left out the information on the determination of p16 and p53 expression. We apologize for this error, and we added a paragraph in the Methods section: “4.4. Flow cytometry for p16 and p53 determination p16 and p53 expression was determined by flow cytometry as described in (Salech et al 2022). Briefly, cCryopreserved PBMCs were resuspended in a PBS solution supplemented with 10% fetal bovine serum, washed with PBS 3% fetal bovine serum (FBS) and fixed using the Cytofix-CytopermTM kit (BD Biosciences). Fixed cells were incubated with anti-human p16 (BD Pharmingen) and anti-human p53 (BD Pharmingen) antibodies for 16 h at 4° C. The secondary antibody was Alexa 488 anti-rabbit (Thermo Fisher Scientific, 1:1000 in PBS 3% FBS for 1 h at room temperature). Quantification was performed using a Cytoflex (Beckman Coulter) flow cytometer and FlowJo software v10.0.10, considering a cell population of 100,000 cells for each analysis.”

  1. line 166: “mRNA expression of p16 (A) and p53 (B) were determined by flow cytometry”. Is the flow cytometry correct?

Yes, as mentioned above p16 and p53 were measured by flow cytometry and we added the description in Methods.  

  1. How p21 and p53 were detected? I cannot find it in Section Materials and methods. Please provide a detailed information.

As mentioned above we added a paragraph in the Methods section explaining the detailed method of p16 and p53 determination.

Round 2

Reviewer 1 Report

The resubmitted article has incorporated some of my comments but still has some questionable aspects. First, the authors are not able to support the current study with a senescence-associated gene map for PBMCs of the patient population.  Though the authors cited the previous gene map studies, unfortunately, it does not justify the present study having patients with aMCI, Ca+aMCI samples. Secondly, this is the continuation of previous work with similar experimental designs. it is a bit confusing for the readers to understand the difference and significance of the studies.  Further H2AX expressions are usually measured as foci/cell. The authors have presented the IF for H2AX as an expression. Given the PBMC with different cell types authors should perform flow cytometry, segregate and perform H2AX staining with the name of the cell type showing the H2AX foci.  The formation of foci is an absolute marker of  DNA DSB formation. it goes exponential in the S phase under stress.  It is not very clear about the cell cycle phase of the IF images presented in figure 3.